# Microtubule-Based Mitochondrial Dynamics as a Valuable Therapeutic Target in Cancer

**DOI:** 10.3390/cancers13225812

**Published:** 2021-11-19

**Authors:** Rosa Vona, Anna Maria Mileo, Paola Matarrese

**Affiliations:** 1Center for Gender-Specific Medicine, Oncology Unit, Istituto Superiore di Sanità, 00161 Rome, Italy; rosa.vona@iss.it; 2Tumor Immunology and Immunotherapy Unit, Istituto di Ricovero e Cura a Carattere Scientifico (IRCCS)—Regina Elena National Cancer Institute, 00144 Rome, Italy

**Keywords:** microtubules, mitochondria dynamics, mitophagy, cancer bioenergetics, tunneling nanotubes

## Abstract

**Simple Summary:**

Mitochondria are well known for being the powerhouses of the cell—whether the cell is normal or cancerous. Moreover, they can move, split, fuse themselves, or be eliminated via mitophagy with the help of the interplay between motor proteins and the cell scaffold—especially microtubules. The relationship between mitochondria, microtubules, and motor proteins is altered in cancer, and targeting this molecular machinery can offer a novel weapon in its treatment. In this paper, we review and summarize the state of the art of this approach.

**Abstract:**

Mitochondria constitute an ever-reorganizing dynamic network that plays a key role in several fundamental cellular functions, including the regulation of metabolism, energy production, calcium homeostasis, production of reactive oxygen species, and programmed cell death. Each of these activities can be found to be impaired in cancer cells. It has been reported that mitochondrial dynamics are actively involved in both tumorigenesis and metabolic plasticity, allowing cancer cells to adapt to unfavorable environmental conditions and, thus, contributing to tumor progression. The mitochondrial dynamics include fusion, fragmentation, intracellular trafficking responsible for redistributing the organelle within the cell, biogenesis, and mitophagy. Although the mitochondrial dynamics are driven by the cytoskeleton—particularly by the microtubules and the microtubule-associated motor proteins dynein and kinesin—the molecular mechanisms regulating these complex processes are not yet fully understood. More recently, an exchange of mitochondria between stromal and cancer cells has also been described. The advantage of mitochondrial transfer in tumor cells results in benefits to cell survival, proliferation, and spreading. Therefore, understanding the molecular mechanisms that regulate mitochondrial trafficking can potentially be important for identifying new molecular targets in cancer therapy to interfere specifically with tumor dissemination processes.

## 1. Introduction

The cytoskeleton is a dynamic and interconnected network of filaments composed of structural and regulatory proteins that play a key role in all fundamental cellular processes, such as shape retention, motility, division, and intracellular transport of proteins and organelles [1,2]. Therefore, it is not surprising that alterations in cytoskeletal function can contribute to the onset and progression of cancer [3]. The three main types of filament that characterize the cytoskeleton are microfilaments, microtubules, and intermediate filaments [4]. Several ultrastructural analyses have shown that the cytoskeletal filaments interact directly or indirectly with the plasma membrane and various intracellular organelles [5].

The microtubules (MTs)—the most rigid intracellular cytoskeletal filaments—are formed by the polymerization of two globular proteins, α- and β-tubulin, into protofilaments that can then associate laterally to form a hollow tube [6].

Microtubules have a distinct polarity that is critical for their biological function. Tubulin polymerizes end-to-end; therefore, in an MT, one end will have the α-subunits (minus) exposed, while the other end will have the β-subunits (plus) exposed [7]. 

MTs are essential in many vital cellular processes, such as structural support, mitosis, chromosome segregation during meiosis, and intracellular transport of vesicles and organelles such as mitochondria [1,2,7]. In particular, to facilitate the movement of vesicles and mitochondria along their tracks, MTs recruit motor proteins via acetylation on lysine 40 of α-tubulin [8,9,10,11]. Microtubule-associated motor proteins include kinesin and dynein, which carry their cargo to the minus and plus ends of the microtubules, respectively [12,13]. In vitro studies have revealed that the loss of acetylated residues in MTs reduces the interaction of kinesin with tubulin, with a subsequent decrease in cell motility [10]. 

MTs have been an ideal target in antineoplastic therapy for many years, as they are the main components of the mitotic spindle. In addition, these filaments, together with motor proteins, play a fundamental role in the mitochondria’s structural and functional organization, including morphology, dynamics, motility, and distribution (Figure 1) [14].

Although the mechanisms regulating this interplay and its impact on mitochondrial architecture and cellular bioenergetics are still not well understood, growing evidence underlines how mitochondrial dynamics are fundamental in tumorigenesis, tumor progression, and the metabolic flexibility of cancer cells [15]. It has been hypothesized that the mitochondria–MT associations are necessary to regulate the distribution, positioning, and tracking of mitochondria to cellular-energy-requiring areas, as suggested in several neuronal studies, where kinesins and dynein were shown to transport mitochondria through axons and dendrites to energy-intensive areas in order to produce adenosine triphosphate (ATP) and guanosine triphosphate (GTP) [12,16]. Furthermore, the interaction of microtubules with the outer membrane proteins’ voltage-dependent anion-selective channel (VDAC) is directly involved in the coordination of mitochondrial function [17]. The intracellular distribution of mitochondria occurs through the action of the motor proteins associated with microtubules, including the plus-end-directed kinesins and minus-end-directed dyneins [18,19]. 

In addition to regulating cell metabolism and energy production, mitochondria play a crucial role in several fundamental cellular activities, including calcium homeostasis, reactive oxygen species (ROS) production, and programmed cell death [20]. Each of these processes can be impaired in cancer cells. The acquisition of migratory and invasive abilities and adaptive changes in the metabolism of cancer cells has often been associated with alterations in the mitochondrial network [21]. Indeed, mitochondrial dynamic processes are key to the maintenance of mitochondrial homeostasis [22]; they include the displacement of mitochondria along the cytoskeleton, and the regulation of mitochondrial architecture mediated by fusion/fission events [22]. Interestingly, in addition to intracellular mitochondrial movement, a horizontal mitochondrial transfer between neighboring or even non-immediately contacting cells was also observed [23]. These exchanges, especially in the cancer microenvironment, can satisfy the energy needs of the acceptor cell, thus favoring its proliferation and survival. 

In this review, we analyze the involvement of the mitochondria–microtubules interplay in tumor progression based on the current knowledge in this field.

## 2. Mitochondria

Mitochondria probably evolved from engulfed prokaryotes that developed an endosymbiotic relationship with the host eukaryote, gradually developing into a mitochondrion [24]. As double-membrane-bound organelles, mitochondria have five distinct compartments: the outer mitochondrial membrane (OMM), the inner membrane space (IMS), the inner mitochondrial membrane (IMM), the cristae (originated from the folds of the inner membrane), and the matrix that contains the mitochondrial DNA [25]. They are considered to be the energy producers of cells, as the cristae host the electron transport chain (ETC) and oxidative phosphorylation (OXPHOS) proteins. Mitochondria are especially located along cell extensions at the anterior edges of cells, where highly energetic mechanisms such as extensive cytoskeletal remodeling and cell adhesion processes occur [26]. Mitochondria play a pleiotropic role in tumorigenesis by allowing cancer cells to adapt to supervening metabolic needs and environmental changes [27]. Recent studies have demonstrated the potential roles of mitochondrial trafficking in cancer cell motility and invasion [28]. 

Mitochondria constitute a dynamic network in continuous reorganization, thanks to the balance between different mechanisms such as fission and fusion, biogenesis, and mitophagy, which control the number, morphology, quality, and cellular distribution of the mitochondria [29]. The mitochondrial dynamics are essential in regulating several cellular functions, playing a crucial role in bioenergetics activities, inflammation, cell differentiation, movement, and cell fate [29].

## 3. Mitochondrial Fission and Fusion

The mitochondrial network morphology continuously changes as a result of fusion/fission processes and the movement of mitochondria along microtubular structures [30]. In particular, the balance between fission and fusion determines the shape, size, and number of mitochondria, strongly impacting on energy metabolism. Emerging evidence indicates that alteration of this balance contributes to various aspects of tumorigenesis, cancer progression, and metastasis.

Fusion and fission are highly energetic cellular processes closely related to the functioning of mitochondrial activity [31]. For instance, the fusion of damaged mitochondria with healthy ones can restore—at least partially—the function of the impaired mitochondria. On the other hand, the fission process allows the segregation of functioning mitochondria from damaged ones, thus enabling the mitophagic removal of the latter [32]. Mitochondrial fusion is a sequential and complex process involving the outer and inner mitochondrial membranes and the matrix. The primary regulators of this process are the GTPase dynamin-related proteins (outer mitochondrial membrane proteins) mitofusin1 (MFN1) and mitofusin2 (MFN2), and optical atrophy 1 (OPA1)—a transmembrane protein tightly associated with the mitochondrial inner membrane, and located in the intermembrane space [31]. 

The opposite process—mitochondrial fission—is mainly regulated by the large GTPase dynamin-related protein DRP1, mitochondrial fission protein 1 (Fis1), and mitochondrial fission factor (MFF) [33], and is responsible for mitochondrial fragmentation. DRP1 is a cytosolic protein, which requires the localization of Fis1 in the mitochondrial outer membrane in order to form the fission complex. DRP1 physically constricts the mitochondrion to form a ring structure located on the future mitochondrial fission area; its phosphorylation regulates the mitochondrial translocation and activation of DRP1 by multiple kinases as a function of the different phases of the cell cycle, or in response to stress conditions [34]. MFF, along with Fis1, appears to be one of the mitochondrial receptors of DRP1 [35]. Accordingly, a reduction in MFF levels induces elongation of the mitochondrial network and a decrease in the translocation of DRP1 to the mitochondria [36]. Recently, the mitochondrial dynamic proteins MID49 and MID51 have been observed to participate in the recruitment of DRP1 to the mitochondria [37]. 

Multiple studies have demonstrated an imbalance of fission and fusion processes in cancer, with elevated fission activity and/or decreased fusion resulting in a fragmented mitochondrial network [33]. Such fragmentation of mitochondria allows their spatial redistribution in cell areas with greater energy needs [38]. It has been proposed that mitochondrial fusion promotes tumor cell resistance to apoptosis, whereas mitochondrial fission has been associated with increased invasiveness. Indeed, several studies have demonstrated that mitochondrial fission is required in order to maintain the migratory and invasion potential of breast, thyroid, and glioblastoma cancer cells [38,39,40,41], while DRP1-induced mitochondrial fission was found to be associated with a migratory phenotype in several types of cancer. In human breast cancer cells, treatment with mitochondrial division inhibitor 1 (MDIVI-1)—a DRP1-specific inhibitor that suppresses mitochondrial fission [42]—induced the re-localization of mitochondria near the nucleus, suggesting inhibition of subcellular mitochondrial trafficking [28]. Notably, recent research has also demonstrated that restoration of the fused mitochondrial network—through either DRP1 knockdown/inhibition or MFN2 overexpression—impairs cancer cell growth, suggesting that mitochondrial network remodeling is essential in cancer progression [38,39,43]. In accordance with the above, a dysregulation of OPA1, MFN1, and MFN2 was observed in different types of human tumors—such as lung and bladder cancers [44,45]—while, in hepatocellular carcinoma, a high expression of DRP1 was associated with a significant increase in distant metastases [46]. All of these facts highlight the important role of mitochondrial dynamics in metastatic processes [33].

In any case, the mechanisms that regulate fission and fusion have not yet been fully identified, but would also seem to be determined by the specific cell type (e.g., yeast, neuron, cardiomyocyte, epithelial cells, etc.). However, in general, it has been observed that mitochondrial motility facilitates fission and fusion, since a mitochondrion moves towards another to merge and, once divided, the mitochondria have to move apart in order to remain separate [47]. In fact, experimental evidence has suggested that impairment of the mitochondrial motility, mediated by nocodazole or vasopressin—causes selective inhibition of the fusion process [48]. 

Previously published data clearly indicate that microtubules play an important role in fusion and fission processes. For example, Mahecic et al. reported that microtubule-based motor proteins were responsible for generating sufficient tension forces to induce the fission process [49]. The actomyosin cytoskeleton participated in the formation of the constriction point, and in the recruitment of DRP1 in the division zone [50,51]. In accordance with this scenario, it was observed that the destruction of microtubules with nocodazole, or of actin filaments with latrunculin-β, inhibited the mitochondrial fission process [51]. On the other hand, it has also been found that the interaction between microtubules and mitochondria via the microtubule–mitochondria binding protein (Mmb1p) could inhibit the localization of DRP1 to the mitochondrion, thus counteracting the fission process [52]. Accordingly, the deletion of Mmb1p induced mitochondrial fission [53]. Mmb1p appears to play a role in the stability of the microtubule network. It has been suggested that more stable microtubules would favor longer contact times between mitochondria and microtubules, thus promoting mitochondrial elongation. Conversely, shorter mitochondria–microtubule interaction times would seem to favor the activation of fission mechanisms, leading to mitochondrial fragmentation [54].

In Figure 2, a schematic drawing summarizing the processes of fission and fusion, and the main actors involved, is shown.

## 4. Mitophagy

Given the crucial role of mitochondria in vital processes, there are several multistep mechanisms involved in the control of their functionality, including mitophagy [30,55,56].

Mitophagy, a specific type of autophagy, is a helpful self-degradative process for mitochondrial quality control [57]; it is critical to clearing damaged or dysfunctional mitochondria and maintaining cellular homeostasis, since dysfunctional mitochondria can promote oxidative stress [58].

The serine/threonine kinase PTEN-induced putative kinase 1 (PINK1), and the E3 ubiquitin ligase Parkin, play pivotal roles in the regulation of mitophagy. PINK1 is normally imported into the mitochondria, where it is cleaved by the protease PARL, and remains in small amounts on the inner membrane. In mitochondrial depolarization conditions with a transmembrane potential (Δψm) decrease, PINK1 levels on the outer membrane increase. Parkin moves from the cytosol to the mitochondria in healthy mitochondria, triggering the ubiquitination of different proteins on the outer membrane, such as MFN1, MFN2, and VDAC. In damaged mitochondria, Parkin is selectively recruited via a PINK1-mediated process [59]. Following ubiquitination, p62/SQSTM1 mediates the interaction between proteins marked by ubiquitin and LC3, allowing the formation of a phagophore able to engulf and degrade the damaged mitochondrion. Parkin-induced mitophagy is dependent on PINK1, but it also requires DRP1-mediated mitochondrial fission [24]. Indeed, fission is critical for mitophagy. In this process, one depolarized and one hyperpolarized mitochondrion are formed, and only the depolarized mitochondrion is removed, whereas the hyperpolarized mitochondrion can be re-introduced into the mitochondrial network [60]. The close association between mitochondrial movement and mitophagy was first indicated by the observation of a biochemical association between PINK1 and the Miro complex [61], and subsequently between Parkin and this complex—especially after mitochondria depolarization with carbonyl cyanide m-chlorophenylhydrazone (CCCP) [62].

As consequence of activating the PINK1/Parkin pathway, there is the proteasome-dependent degradation of Miro and the subsequent release of kinesin from the mitochondrial surface [62,63]. All of this determines the arrest of mitochondrial transport and the recruitment of cytosolic Parkin to the mitochondrion [62]. It is therefore likely that halting mitochondria in some manner facilitates their clearance by mitophagy.

An alternative pathway for the induction of mitophagy—particularly important in cancer cells—is activated by hypoxia. Damaged mitochondria increase the expression of BNIP3, BNIP3-like (BNIP3L/NIX), and FUNDC1—a family of mitophagy receptors localized in the OMM of the mitochondria [64], which directly recruit LC3 through their LC3-interacting region (LIR) to initiate mitophagy [65,66]. BNIP3 and NIX interact with LC3 at the microtubule level, promoting the sequestration of mitochondria in forming autophagosomes [67]. Figure 3 shows both alternative pathways. 

A close link has been observed between mitophagy and microtubules in aggressive tumors, such as glioblastomas and metastatic melanomas. In particular, in a model of glioblastoma, a reduction in α-tubulin has been observed to induce a downregulation of BNIP3 and NIX, with consequent inhibition of mitophagy. This leads to a reduction in the numbers of of lamellipodia and filopodia, with a significant reduction in the migratory capacity of tumor cells [68]. 

Mitophagy is also a crucial complex process in the progression of hematological malignancies and the acquisition of drug resistance, especially in advanced myeloma and lymphomas [69]. In high-grade lymphomas and in the cells derived from particularly aggressive tumors, the fusion between mitophagosomes and lysosomes frequently occurs in the perinuclear zone, at the minus end of the microtubule network [70]. In these cells, the mitochondrial localization around the nucleus is strongly fission-dependent [71]. DRP1 and Fis1 are master regulators of fission machinery, and act in the asymmetric cell division of stem cells, facilitating the preservation of stem properties only to daughter cells that inherit the younger mitochondria [72]. 

It is interesting to note that mitophagy can play opposite roles in tumorigenesis, based on the tumor type and stage and the microenvironmental context. Indeed, this process can promote the survival of cancer cells by eliminating damaged mitochondria that, through excessive ROS production, could induce apoptosis. At the same time, mitophagy can act as a tumor suppressor by eliminating impaired mitochondria that, inducing a chronic mild oxidative stress, could promote carcinogenesis. In general, in the first steps of carcinogenesis, Parkin mutations inhibit mitophagy, while during cancer progression, abnormal regulation of BNIP3 improves mitophagy. This adaptation process may represent a cellular strategy for increasing cancer survival [73]. For instance, it has been demonstrated that in the onset of hepatocellular carcinoma, the loss of mitophagy induces the accumulation of damaged mitochondria, promoting carcinogenesis [74].

It should be noted that alterations in mitochondrial dynamics and mitophagy are considered to be among the most important causes of mitochondrial DNA (mtDNA) release [75]. Cytosolic mtDNA fragments can translocate into the nucleus and be incorporated within nuclear DNA, contributing to genomic instability and potentially causing cancer and other diseases [76]. Interestingly, cytosolic mtDNA is a potent agonist of the cell’s innate immune surveillance machinery; it can trigger an innate inflammatory response [77], enabling the recruitment of adaptor molecules/receptors—such as cyclic GMP–AMP (cGAMP) synthetase (cGAS), toll-like receptor 9 (TLR9), and the nucleotide-binding oligomerization domain-like receptor family pyrin domain-containing 3 (NLRP3) inflammasome—which induce a type I interferon (IFN-I)- or NF-κB-mediated inflammatory response [77,78,79].

Interestingly, in a recent paper, Ziegler et al. highlighted a link between mitophagy, lysosomal integrity, and MHC class I presentation in intestinal epithelial cells (IECs). In particular, the authors demonstrated the active role of the immune system in the antitumor response in colon cancer, supporting the possibility of successfully modulating the immune response in at least some types of cancer. The hypothesis arising from these results is that the therapeutic trigger of mitophagy could stimulate antigen presentation in the tumor cells themselves, contributing to the development of an immune response against colorectal cancer [80]. 

## 5. Intracellular Mitochondrial Trafficking

A fundamental step in tumor progression that improves invasiveness and metastatic propensity is the motility increase of cancer cells. Growth factors and cytokines regulate the cell migration process through different signaling pathways—such as MAPK and PI3K-AKT—which alter the expression of genes involved in cell polarity, morphology, cytoskeletal dynamics, and cell adhesion, increasing migratory ability [81]. The importance of the spatial distribution of mitochondria in cancer cells, and the mechanisms by which mitochondrial dynamics regulate cell migration, have only recently been brought to light. Mitochondrial trafficking has emerged as a fundamental regulator of the metastatic capacity of various tumors [26]. Indeed, our current knowledge shows that the localization of mitochondria to the leading edge favors tumor invasion by providing the ATP and metabolic intermediates necessary for the bioenergetic and biosynthetic demands of the cells. A high amount of energy is needed to power the cytoskeletal dynamics and the different molecular processes, such as the development of focal adhesions and cell protrusions essential for cell migration [26]. A recent study showed that cortical mitochondria supported membrane lamellipodia dynamics and actin cytoskeleton remodeling, resulting in increased cancer cell motility and invasion [82]. The importance of local energy production was demonstrated in both ovarian cancer cells [83] and living mouse embryonic fibroblasts (MEFs), where mitochondrial accumulation at the leading edge of the lamellipodia led to increased ATP concentration [84].

Mitochondrial localization in cancer cells can be reprogramed depending on intracellular and extracellular signals, leading to cells changing from a highly proliferative phenotype to a highly invasive phenotype. In particular, the presence of abundant perinuclear mitochondria characterizes a highly proliferative phenotype, while mitochondrial localization to the leading edge determines a highly invasive phenotype (Figure 4A). Thus, mitochondrial re-localization at the cortical level involves a “regional” increase in oxidative metabolism to support the energy-intensive movements [84] and, in general, contributes significantly to the metabolic plasticity of cancer cells. 

The intracellular localization of mitochondria is the result of movements along the microtubules and anchoring to the actin filaments [9]. Protein adapters and mitochondrial receptors make the binding between mitochondria and motor proteins possible. The interaction between motor proteins, adapters, and receptors ensures targeted movements of the mitochondria and the fine-tuning of their motility [85,86]. The molecular mechanisms underlying this movement were initially described in neurons, where microtubule polarity and structural organization influence both soma-to-axon and soma-to-dendrite mitochondrial transport. Microtubule-based motor proteins such as the kinesin superfamily proteins and cytoplasmic dyneins sustain long-range mitochondrial transportation in the anterograde (microtubule plus end) and retrograde (microtubule minus end) directions, respectively. In the axonal portion, the microtubules are uniformly distributed, so that the negative ends face the cell body while their positive ends point distally [87,88]. Although initially considered “neuronal-specific,” the anterograde (from the nuclei to the periphery) and retrograde (from the periphery to the nuclei) mitochondrial movements have also been shown in other cell types, such as migrating lymphocytes [89] and tumor cells [90]. Previously published data demonstrate that the intracellular transport of mitochondria occurs mainly via the microtubule cytoskeleton, using a mechanism consisting of mitochondrial Rho GTPases (MIRO 1/2), trafficking adapter proteins that bind to kinesin (TRAK1 and TRAK2) and the motor proteins kinesin-1/3 and dynein [91,92]. 

The scaffolding TRAK1/2 proteins permit mitochondrial motility by coordinating the interaction between kinesins/dyneins [65] and the Ca^2+^-dependent MIRO GTPase placed on the outer mitochondrial membrane (Figure 4B, left panel) [92]. 

Interestingly, the experimental findings of Heindrichs et al. demonstrated that TRAK1 strongly increases KIF5B’s processivity when the microtubule surface is crowded with a large variety of proteins; moreover, the authors suggest that the anchoring of KIF5B by TRAK1 increases the time for which KIF5 can stop in front of an obstacle without detaching from the microtubule [93].

In contrast, short-range mitochondrial movements depend on actin filaments and myosin motors (e.g., MYO19, MYO6, MYO5). Myosins move along actin filaments in both directions [91]. How myosins regulate movement, and how they bind to mitochondria, is poorly understood. Recently, the MIRO-dependent localization of MYO19 to the mitochondria has suggested that MIRO proteins might be active in regulating mitochondrial motility via either actin or microtubules (Figure 4B, right panel) [94].

Mitochondrial trafficking was first thought of in neurons as an energy supply process toward high-consuming sites [9]. However, it can also locally fuel membrane dynamics and migration of cancer cells [82]. By exploiting the same neuronal regulators of mitochondrial motility, cancer cells can reposition the mitochondria in cortical areas favoring invasive processes [95]. 

The activity of DRP1 appears to be mandatory in the mitochondrial trafficking associated with tumor chemotaxis [86], as mitochondrial fission allows for a more rapid transfer of mitochondria along the microtubules within tumor cells. Consequently, the occurrence of a link between microtubule-based mitochondrial trafficking and mitochondrial fission was suggested [96]. 

It is interesting to note that several mechanisms can determine the blocking of mitochondrial movement and, more generally, the movement of all intracellular organelles inside the cell. Mitochondria can be immobilized (1) by the binding of myosin to actin [96]; (2) by their anchor to microtubules via syntaphilin (SNPH) [97]; (3) by the action of calcium on microtubules [98]; and (4) by proteasomal degradation of the kinesin-1/TRAK complex (Figure 4C) [98].

As the intracellular distribution of mitochondria can regulate tumor cell growth, motility, and metastatic capacity, the alteration of mitochondrial movement could modify cancer therapy responses. Blocking mitochondrial movement would result in a lower energy supply for cancer cells, thus preventing tumor progression and invasion. It has been shown that SNPH can block invasion in glioblastoma, as well as breast, lung, and prostate cancers [95]. Furthermore, lower levels of SNPH are correlated with tumor progression and metastatic dissemination in lung, colon, prostate, and breast cancers [95].

Changes in the intracellular levels of ROS are also able to regulate mitochondrial dynamics. Indeed, several in vitro and in vivo studies on cancer cells have reported that increased ROS production was correlated with mitochondrial membrane potential loss, mitochondrial fission, mitophagy, and apoptosis [99,100]. On the other hand, excessive fission activity can enhance ROS production [101], due to mitochondrial membrane depolarization [102]. In turn, ROS induce post-translational modifications of DRP1, MFNs, and OPA-1, with consequent damage to mitochondrial morphology and function [103]. On the other hand, lowering ROS levels leads to mitochondrial fusion [102].

Thus, the activity of ROS might be capable of increasing tumorigenesis and/or promoting cancer progression by activating signaling pathways that regulate cellular proliferation, metabolic adaptation, apoptosis resistance, chemoresistance, and cellular migration/invasion [101]. 

## 6. Role of Microtubules in Mitochondrial Dynamics

The movement of mitochondria along MT tracks is regulated by second messengers generated ad hoc. Within the past decade, experimental evidence has shown the key role of calcium in regulating mitochondrial movement. High calcium concentrations have been observed in many cell types to inhibit MT-based mitochondrial trafficking by binding to the MIRO1 and 2 proteins [13,16,104]. This link between calcium and MIRO prevents the latter from interacting with the motor protein KIF5 [105] (Figure 4C). With calcium being the second messenger of a plethora of signaling pathways, mitochondrial trafficking can therefore be regulated by many factors [105].

Mitochondria, along with other organelles, constitute intracellular storage sites for calcium. Therefore, it was hypothesized that mitochondrial trafficking could be inhibited or stimulated by calcium fluctuations rather than by the absolute level of calcium [106]. From this perspective, the MIRO/KIF5 binding could represent an indicator of high local calcium levels, allowing the mitochondria to buffer it. The calcium fluxes occur in areas of high metabolic demand, such as nerve endings, or the protrusion zones and leading edge in the case of cancer cells. These areas where the mitochondria are clustered represent the cell migration fronts, and play a pro-metastatic role [107].

In addition, mitochondrial trafficking can also be controlled by the ubiquitination of SNPH or MIRO1. For instance, it has been shown that the ubiquitination of some residues of SNPH—a protein located in the OMM [108]—is necessary to allow binding with tubulin and the consequent relocation of mitochondria to specific cellular areas [105]. By contrast, MIRO1 degradation induces mitochondrial arrest movements due to its phosphorylation at S156 by PINK1. In tumor cells, SNPH is downregulated by oxidative stress. During oxidative stress or hypoxia, the downregulation of SNPH, acting on the mitochondrial metabolism and trafficking, could inhibit cell proliferation and stimulate the motility and invasion of tumor cells. For instance, the degradation of SNPH in hypoxic conditions induced a greater presence of cortical mitochondria in glioblastoma cells, with a consequent increase in their invasiveness [109]. Therefore, SNPH could function as a metastatic propensity regulator, thus proving to be potentially useful as a biomarker. This hypothesis would also agree with the lower levels of SNPH found in cells isolated from metastatic sites compared to those isolated from their respective primary sites.

Other important modulators of mitochondrial dynamics are the ROS that suppress mitochondrial motility in both Ca^2+^-dependent and -independent manners [110,111]

A recent work has shown how ROS could also regulate mitochondrial dynamics via the MAPKp38 pathway (Figure 4C). In particular, in human fibroblasts, a high level of ROS production was able to activate p38, which promoted disengagement of the motor from the microtubule tracks via phosphorylation of the serine residue at position 176 of KIF5 [111]. This inhibited the mitochondrial motility independently of any changes in calcium flux. 

Moreover, it was shown that in neuronal cells under hypoxic conditions, the MIRO/TRAK complex regulated mitochondrial trafficking via its association with hypoxia-upregulated mitochondrial movement receptor (HUMMR) [112].

## 7. Metabolic and Phenotypic Consequences of Mitochondrial Transfer

Multiple studies have shown that whole functional mitochondria can be naturally transferred from a healthy cell to a recipient cell via nanotubular structures known as “tunneling nanotubes” (TNTs) (Figure 5) [113]. TNTs are short-lived cytoplasmic bridges between cells that transport various cargos in a uni- or bidirectional fashion—including cytosolic molecules, organelles such as mitochondria [114,115], or pathogens [116]. TNTs are ultrafine and very heterogeneous in length and width; they lack any attachment to the substrate, but their structure—depending on the context and the delivered cargo—is supported by cytoskeletal F-actin fibers [113] in conjunction with microtubules [117,118]. The main molecular mechanisms driving TNT formation start from the formation of membrane protrusions (filopodia-like) or the dislodgement of two previously attached cells, in both physiological and pathological environments. Each of these processes of cell-to-cell communication can lead to closed-ended or open-ended TNTs, the latter allowing cytoplasmic continuity between connected cells. The TNT-mediated intercellular transfer can occur between neighboring cells or cells not immediately in contact; it may affect the bioenergetic state of acceptor cells, depending on their metabolic requirements to favor cell proliferation and survival [119], resulting in metabolic reprogramming of connected cells [114,120]. In particular, the experimental findings of Tan et al. showed that the transfer of mtDNA from host cells to tumor cells with compromised respiratory function restores the mitochondrial respiration required for tumorigenesis in murine lung and breast tumor models [121]. These results are also supported by the recent data obtained by Bajzikova et al., which confirm the importance of mtDNA transfer from host cells to tumor cells in the reconstitution of OXPHOS, showing that pyrimidine biosynthesis dependent on respiration-linked dihydroorotate dehydrogenase (DHODH) is necessary for tumor growth, and that mitochondrial ATP generation is actually unessential for tumorigenesis [122].

For efficient mitochondrial shuttling, TNTs are formed de novo; they are transiently expressed in response to a broad range of cellular stressors [123,124,125,126,127], suggesting that TNT formation may represent a type of stress response [128]. TNT structures involved in mitochondrial transfer were observed as a heterotypic connection between non-malignant and cancer cells in many different cancer types [129,130], as well as from mesenchymal stem cells (MSCs) to differentiated cells, in damaged tissues and tumors [131]. The ability of TNTs to form between tumor cells and, at the same time, to connect these cells to the tumor microenvironment (TME), indicates a crucial role of mitochondrial trafficking in cancer progression. It has been demonstrated that tumor cells can employ mitochondrial transfer to modify their microenvironment, thus favoring tumor progression [132]. TNT-mediated acquisition of healthy mitochondria confers more aggressive phenotypic characteristics to tumor cells, such as enhanced proliferative and invasive properties and radio/chemotherapy resistance [133,134].

In tumor cells, the advantage of mitochondrial transfer benefits cell proliferation and survival, increases OXPHOS and, consequently, supports cancer metabolic plasticity [130,135]. On the other hand, restoration of basic mitochondrial activities in cancer cells via uptake of healthy mitochondria led to a significant decrease in intracellular ROS levels, suggesting a crucial role for these reactive molecules in the acquisition of chemoresistance after mitochondrial transfer [119].

## 8. Mitochondrial Dynamics and Cancer Therapy

The fundamental role of mitochondria in the different stages of carcinogenesis and in tumor maintenance has led many researchers to hypothesize that mitochondrial dynamics may represent a possible innovative therapeutic target [42,136,137]. 

However, before this can be realized, in-depth studies are necessary in order to shed light on some contradictions emerging from the studies carried out to date. 

For instance, several experimental data have highlighted the dual role of mitophagy in the onset of cancer, based on the type and stage of the tumor and the microenvironmental context. In fact, mitophagy can promote cancer cell survival by removing damaged mitochondria, thus counteracting ROS-mediated apoptosis. On the other hand, mitophagy can act as a tumor suppressor by eliminating dysfunctional mitochondria able to promote carcinogenesis by inducing a mild chronic oxidative stress [73,138].

In aggressive tumors, such as glioblastomas and metastatic melanomas, a close link between mitophagy and tubulin alterations has been observed. In particular, in a model of glioblastoma, the α-tubulin decrease—due to genetic alteration or pharmacological treatment—induced a downregulation of BNIP3 and NIX, and inhibited the selective mitophagic removal of mitochondria. This inhibition of mitophagy resulted in decreased formation of lamellipodia and filopodia able to negatively affect tumor cell migratory ability [64,65]. 

As mentioned above, the highly dynamic network of mitochondria is preserved by the continuous balance between fission and fusion, which are regulated, among others, by DRP1and MFNs, and OPA1, respectively. 

Although mitochondrial fusion has been correlated with chemoresistance in some cancers, most of the literature agrees that the DRP1-induced fission is necessary for the processes of invasion and metastasis in tumors such as those of the breast and thyroid, as well as in glioblastoma [33,38,39,40,41,42]. In accordance with this, in cancer cells a surplus of fission is generally caused by upregulation of DRP1 expression, leading to the formation of fragmented mitochondria necessary for their spatial redistribution to those regions of the cell with high metabolic demands [37]. Given that DRP1 upregulation is a common event in many oncogenic transformations, it can be assumed that cancer cells may be preferentially sensitive to DRP1 inhibition. This hypothesis was confirmed via the pharmacological and genetic inhibition of DRP1, which led to decreases in the growth of glioblastomas, melanomas, hepatocellular carcinomas, and mesotheliomas, either in vitro or in vivo [137,139,140,141]. In the MDA-MB-231 and MDA-MB-436 breast cancer cell lines, the downregulation of DRP1 or overexpression of MFNs had a similar impact in reducing cell migration and invasion. This could suggest that the inhibition of fission may have the same effect as the induction of fusion, at least in some cancers, pointing to the role of mitochondrial dynamics, rather than fission, in the metastatic process [38]. In the same vein, the observed imbalance of the fusion/fission process (i.e., with a predominance of fission) in human lung cancer cell lines could be reversed by DRP1 inhibition (or MFN2 overexpression), promoting cell cycle arrest and increasing spontaneous apoptosis [50]. Furthermore, in brain tumor cells, the inhibition of DRP1 has been reported to decrease migration and proliferation [137]. 

Zhao et al. also showed that mitochondrial fission was necessary for the redistribution of mitochondria to the leading edge, and that this presence enhanced the formation of lamellipodia. The mitochondrial clustering in the migration front of the cell could represent a prerequisite, or be the first step, in the migration and invasion of breast cancer cells [38]. In addition, some studies also support the idea that the inhibition of mitochondrial fragmentation might represent a useful therapeutic strategy to reduce metastatic dissemination in colon cancer cells, in which DRP1 downregulation decreased proliferation and increased apoptosis [136]. 

Although no specific inhibitors targeting MFNs and OPA1 have been devised at now, the hydrazone M1, which acts as a mitochondrial fusion process promoter independently of these two proteins, might be considered a promising drug for targeted cancer therapy [142]. 

Conversely, two drugs inhibiting DRP1 have been developed, i.e., the mitochondrial division inhibitor MDIVI-1, and the peptide P110. The former inhibits DRP1 activity [39], while the latter alters the DRP1–Fis1 interplay, decreasing DRP1’s functionality in the neurons [143]. Between the two agents, MDIVI-1 has been extensively studied in a cancer setting and, although it has shown cytoprotective effects in non-transformed cells—such as neurons and cardiomyocytes—it has shown some cytotoxic properties across a wide range of cancer cell lines [144], thus suggesting a certain selectivity. Moreover, a recent study indicated that MDIVI-1, in addition to inhibiting DRP1, was also able to target mitochondrial complex I in the absence of DRP1, thus directly impacting mitochondrial metabolism [145]. These data further support the role of DRP1 as putative target of pharmacological approaches aimed at inhibiting oncogenic transformations in a wide range of cancers [137,139,140,141].

Inhibition of DRP1 by MDIVI-1 has also been observed to promote apoptosis induced by the cytokine tumor-necrosis-factor-related apoptosis-inducing ligand (TRAIL) in human ovarian cancer cells [146]. TRAIL is a receptor-mediated inducer of apoptosis proposed for the clinical therapy of some cancers, such as pancreatic cancer, non-squamous non-small-cell lung cancer, and lymphoma [147,148]; however, as with most drugs, the resistance acquired by tumor cells limits their therapeutic effectiveness over time. Similarly, MDIVI-1 was found to be active in overcoming cisplatin resistance in primary ovarian cancer cells isolated from patients [149]. The inhibition of mitochondrial fission would therefore seem to sensitize tumor cells to antineoplastic drugs, suggesting a possible use of MDIVI-1 in combined therapy. Interestingly, in cardiovascular diseases, the inhibition of the mitochondrial fusion process has been suggested to represent a promising therapeutic strategy. In fact, MFN1- and -2-deficient cells were characterized by elevated mitochondrial fragmentation with a loss of mitochondrial membrane potential and defects in mitochondrial respiration [141,150]. Ferreira et al. demonstrated that in rats’ heart failure, β-II protein kinase C (βIIPKC) accumulates on the mitochondrial outer membrane and phosphorylates MFN1, resulting in buildup of fragmented and dysfunctional mitochondria. The authors showed that the use of βIIPKC siRNA or a synthetic βIIPKC inhibitor mitigated mitochondrial fragmentation and cell death in cultured neonatal and adult cardiac myocytes [150]. 

As emerged from the above, the localization of mitochondria in the different areas of the cell strongly impacts its proliferative and movement capacities and, therefore, plays a fundamental role in the spreading of tumor cells.

In this regard, although initially described as neuronal-specific, SNPH is expressed in multiple non-neuronal tissues, including cancers [26,109]. A decrease in SNPH causes a considerable mitochondrial repositioning to the cortical cytoskeleton, enhancing cancer cell motility and invasion. It was demonstrated that SNPH downregulation or loss during tumor progression was correlated with poor outcomes in patients [109]. Conversely, the reintroduction of SNPH into invasive tumor cells was able to decrease metastatic dissemination in a murine model [151].

Given the role played by the binding between the mitochondria and the cytoskeleton in the regulation of mitochondrial dynamics, microtubule-targeted agents constitute a class of anticancer drugs used in the clinic [38,152]. Among the most widely used agents in the treatment of several malignancies, there are taxanes and vinca alkaloids [84,153]; their use is mainly justified by the fact that, by interfering with the formation of the mitotic spindle, they have an antiproliferative effect. However, we cannot exclude the possibility that their anticancer efficacy is also partly linked to the effect exerted on the mitochondrial dynamics. 

Targeting DRP1, SNPH, or other proteins involved in mitochondrial dynamics could therefore be of great interest in the context of anti-metastatic therapy. In fact, although metastases are the leading cause of death in cancer patients, there is a scarcity of therapeutic targets to interfere specifically with tumor dissemination processes [154]. 

In accordance with the growing evidence of the contribution offered by mitochondrial dynamics in metastatic processes—promoting both metabolic adaptation and the migration propensity of cancer cells [155,156]—the biochemical machinery involved in these dynamics may represent an innovative therapeutic target.

## 9. Conclusions

In recent decades, it has emerged that dynamic interactions between mitochondria and the cytoskeleton are critically important for maintaining the structure and function of the mitochondrial network. The movement of mitochondria through the cytoskeleton is fundamental for the supply of energy and metabolites to areas of the cell with high energy demands, and for buffering calcium where necessary. Furthermore, the cytoskeletal network—particularly microtubules and motor proteins—plays a fundamental role in the regulation of the mitochondrial fission/fusion balance, as well as in quality control, mitochondrial turnover, and in the distribution of mitochondria during cell division. 

Since cancer is a disease associated with mitochondrial dysfunction, which has a key role in carcinogenesis, as well as in tumor maintenance and progression, considering mitochondrial dynamics as an innovative therapeutic target and/or as a useful prognostic biomarker in cancer might be appropriate. In this scenario, further studies are needed in order to better understand the effects of different oncogenic signaling pathways on mitochondrial dynamics, and/or to identify additional signaling modalities that regulate mitochondrial network homeostasis in cancer cells—also as a function of the tumor microenvironmental features.

## Figures and Tables

**Figure 1 cancers-13-05812-f001:**
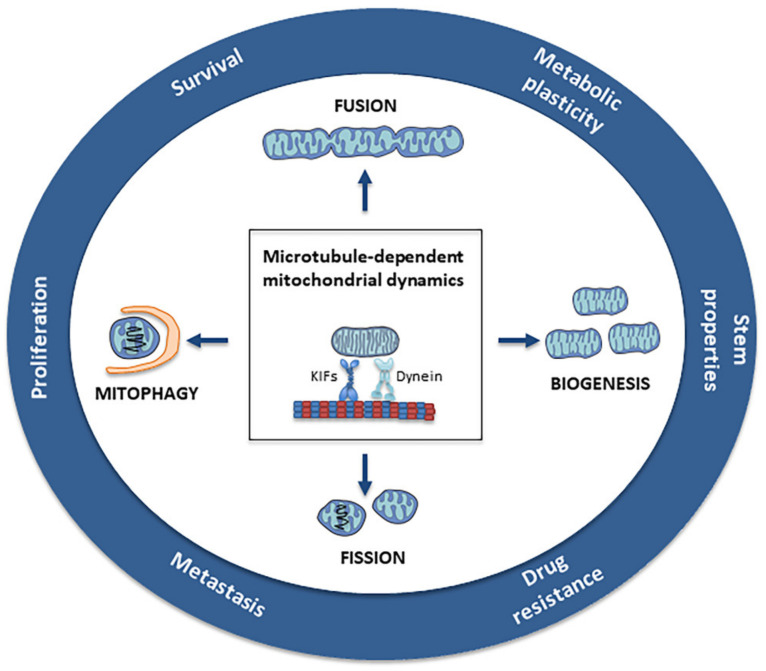
Microtubule-dependent mitochondrial dynamics: Through the balance between fusion/fission and biogenesis/mitophagy, mitochondrial dynamics represent a central process in the bioenergetic adaptation and metabolic plasticity of cancer cells. The balance between biogenesis and mitophagy regulates the number of mitochondria and their quality. The fusion process helps to increase mitochondrial metabolism and to limit mitophagy and apoptosis, while the fission process allows the spatial redistribution of mitochondria in areas of the cell with greater energy and metabolic needs, favoring cell spreading and metastases.

**Figure 2 cancers-13-05812-f002:**
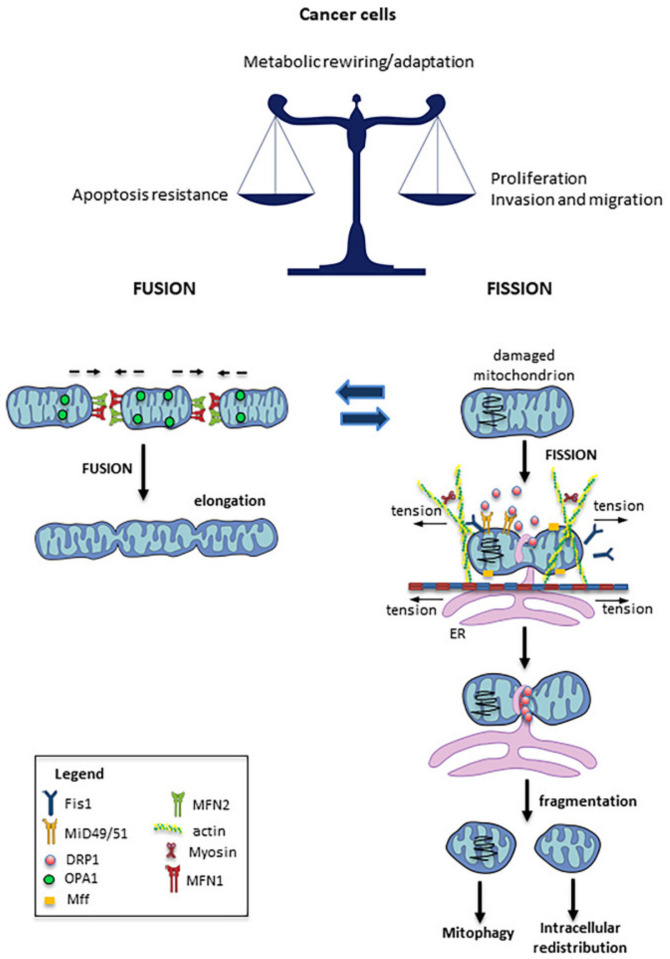
Mitochondrial fission and fusion: Mitochondrial fusion is mainly regulated by MFN1, MFN2, and OPA1 activity, which promote the fusion of juxtaposed mitochondrial membranes. The fusion process contributes to implementing respiration and mitochondrial metabolism, while limiting mitophagy and apoptosis. Mitochondrial fission is regulated by the GTPase activity of the DRP1 that is recruited to the mitochondria in response to stresses, and here interacts with its mitochondrial receptors (Mff1, Fis1, and MID49/51). DRP1 is responsible for mitochondrial fragmentation, as it physically constricts the mitochondrion by forming a ring structure located on the future mitochondrial fission area.

**Figure 3 cancers-13-05812-f003:**
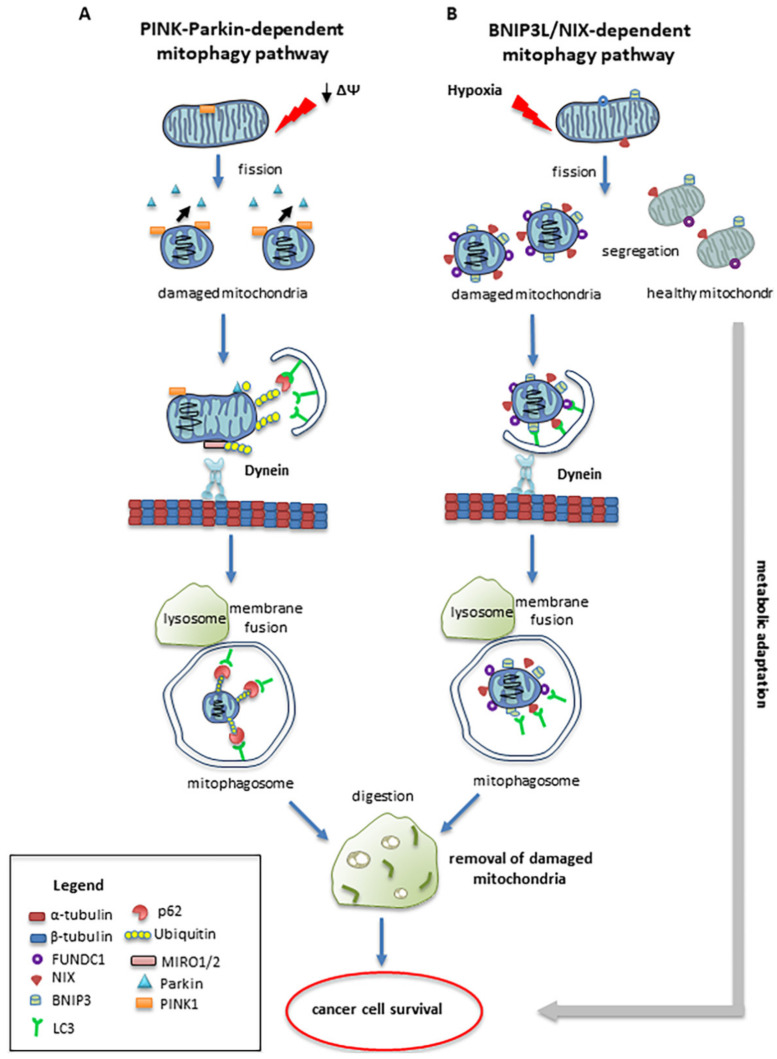
Mitophagy: Mitophagy is a specialized form of autophagy in which dysfunctional mitochondria are targeted and engulfed by autophagosomes that fuse with lysosomes to degrade the encapsulated mitochondria. Mitophagy is regulated by a number of different mechanisms, including Pink1/Parkin-mediated pathways and the BNIP3/NIX pathways. (**A**) When mitochondria are damaged by losing their membrane potential (ΔΨ), PINK1 recruits Parkin from the cytosol to the damaged mitochondria. Here, phosphorylated Parkin ubiquitinates outer membrane mitochondrial proteins, and causes mitochondrial engulfment by binding to LC3 on the isolation membranes that fuse with lysosomes. (**B**) BNIP3/NIX-mediated mitophagic pathways are activated in cancer cells by hypoxia. Outer mitochondrial membrane proteins, such as BNIP3/NIX, bind to LC3 on the isolated membranes, mediating the sequestration of damaged mitochondria into autophagosomes.

**Figure 4 cancers-13-05812-f004:**
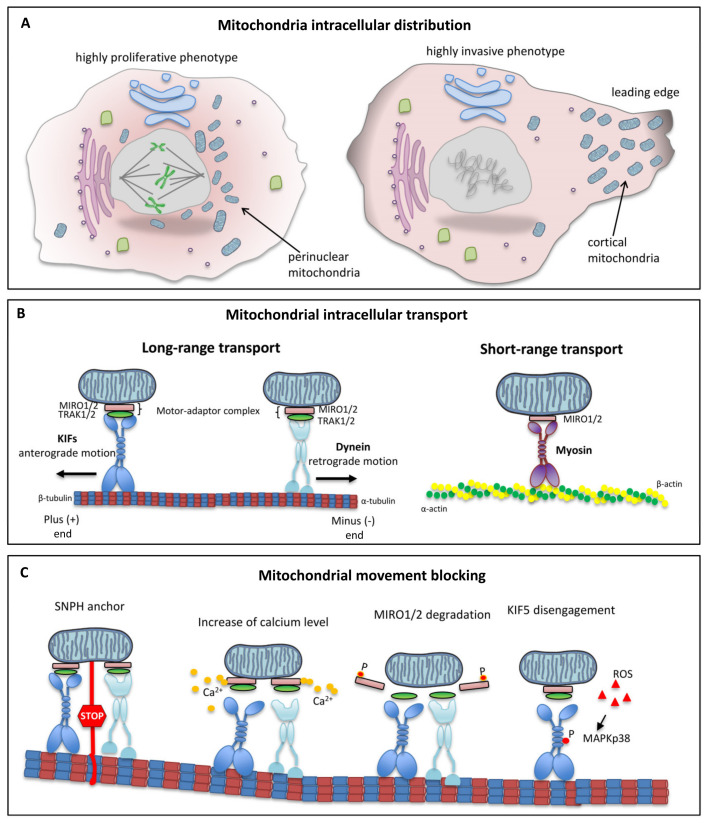
Intracellular mitochondrial trafficking: (**A**) Mitochondrial intracellular distribution. In cancer cells, the presence of abundant perinuclear mitochondria characterizes a highly proliferative phenotype; mitochondrial localization to the leading edge determines a highly invasive phenotype. (**B**) Mitochondrial intracellular transport. Mitochondria move within the cell over short and long distances. Microtubule-based mechanisms drive long-distance transport, while short-distance transport is driven by actin-based movement. Long-distance transport is performed by two MT-based molecular motors with opposite functions: the kinesin (KIF)-driven anterograde transport, and the dynein-driven retrograde transport. (**C**) Mitochondrial movement blocking. Mitochondria possess several anchoring mechanisms capable of blocking their movement. SNPH associates with the mitochondrial outer membrane and anchors mitochondria to microtubules. Moreover, high calcium concentrations inhibit MT-based mitochondrial trafficking by binding to the MIRO 1/2 proteins. This binding prevents MIRO and KIF from interacting. In addition, mitochondrial trafficking can also be controlled by the ubiquitination of SNPH or MIRO 1/2, or by high levels of ROS production able to activate MAPK p38. The p38 phosphorylation promotes disengagement of the KIF from microtubule tracks via phosphorylation of serine 176.

**Figure 5 cancers-13-05812-f005:**
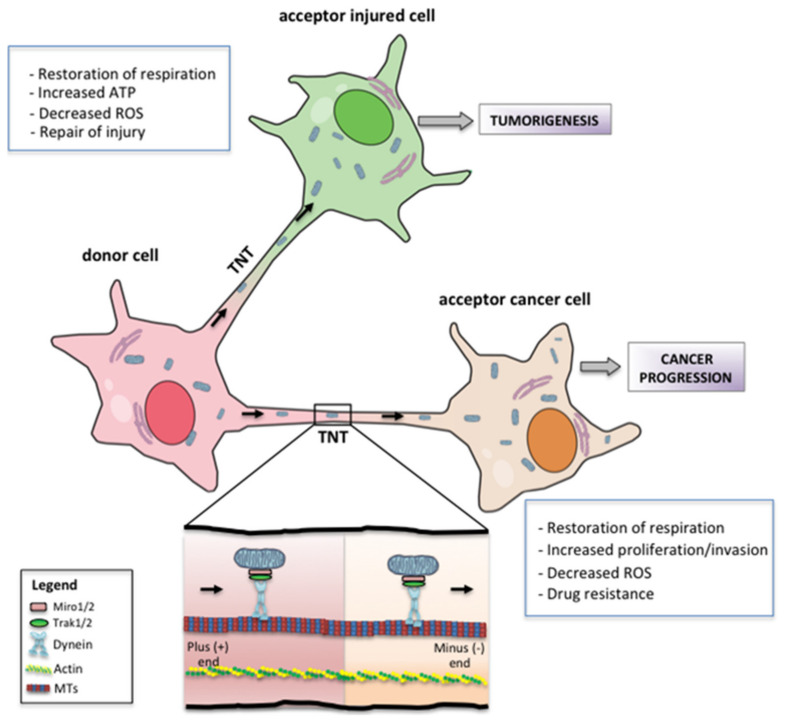
Mitochondrial transfer: Schematic representation of mitochondrial transfer via tunneling nanotubes (TNTs). The donor cell—generally a non-cancerous cell—moves mitochondria to the recipient cell, usually a cancer cell or an injured cell. The main advantages of mitochondrial transfer to acceptor cells are listed.

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
