# Peer review of "Microtubule-Based Mitochondrial Dynamics as a Valuable Therapeutic Target in Cancer"

_cancers, 2021, doi:10.3390/cancers13225812_

Round 1
Reviewer 1 Report
In this manuscript, Vona, Mileo and Matarrese review the role of mitochondria in cancer and how microtubules might localize mitochondria to benefit cancer cells. The review provided an interesting overview of this subject. However, there are two major concerns. First, the legends for the figures are minimal; the legends should describe functions of the components depicted and the processes depicted. This information is, in some instances, obscure or missing in the text. The second major concern is the References section, which should be carefully revised. While citing reviews is appropriate for general concepts, research articles should be cited for specific data. One example of many is, “… recent research conducted by Senft and Ronai also demonstrated…”, citing 32, which is actually a review. Citations should also be correct and accurate. For example, references 7-9 are used for recruitment of motor proteins by alpha-tubulin K40 acetylation. References 7 and 8 have nothing to do with this information, and 9 is a review. This information should have been cited as Reed et al. Curr Biol. 2006. Finally for the references, some of the older reviews should be replaced with more current/updated versions.
Some other points are as follows:
The first sentence of the Introduction: transcription factors are proteins.
There probably should be a reference for the sentence ending on line 46.
There probably should be a reference for the sentence ending on line 55.
Line 57, “Dynein has been reported…” is outside the subject of the paragraph.
Line 101, process to processes.
Reference for sentence ending on line 115.
The sentence starting on line 119 is confusing due to the parathesis (missing one); the next sentence is outside the subject of the paragraph, and it is unclear why the authors focused on MFN2 (why not OPA1?).
Line 134, “MFF appears to be the mitochondrial receptor for DRP1.” It is one of the receptors, along with Fis1 (should probable cite Loson et al. Mol Biol Cell. 2013.
Line 194, clearance damaged to clear damaged.
Line 201, from cytosol to from the cytosol.
Line 212, OMM introduced for outer mitochondrial membrane was also introduced on line 95 but not used consistently throughout the manuscript.
Line 337, several experimental evidence to experimental evidence
Lines 448-449, “or to be” to or be
There are examples when mechanistic information is stated without describing its overall significance/relevance. For example, what is the significance of perinuclear mitophagy in lymphomas? Which process at the leading edge of migrating cells require energy? Current knowledge of mechanisms controlling anterograde versus retrograde mitochondrial transport?
Author Response
Replies to the comments
Reviewer #1
We are glad to have received the fruitful comments and suggestions that allowed us to make substantial improvements to our work.
We modified the manuscript according your remarks. You can find a point-by-point reply to your comments below.
1) Reviewer: Author: the legends for the figures are minimal; the legends should describe functions of the components depicted and the processes depicted.
Authors: Thank you for your remark. We modified figure legends as suggested.
2) Reviewer: The second major concern is the References section, which should be carefully revised. While citing reviews is appropriate for general concepts, research articles should be cited for specific data…. Citations should also be correct and accurate. For example, references 7-9 are used for recruitment of motor proteins by alpha-tubulin K40 acetylation. References 7 and 8 have nothing to do with this information, and 9 is a review. This information should have been cited as Reed et al. Curr Biol. 2006. Finally for the references, some of the older reviews should be replaced with more current/updated versions.
Authors: We apologize for these inaccuracies, which, in the revised version of the paper, have been corrected. Furthermore, some older references have been replaced by newer ones and only those old references that can be considered milestones have been kept.
3) Reviewer: The first sentence of the Introduction: transcription factors are proteins.
Authors: We thank the referee for his/her suggestion. In the revised version the sentence has been modified.
4) Reviewer: There probably should be a reference for the sentence ending on line 46.
Authors: We thank the referee for his/her suggestion. In the revised version the reference has been added.
5) Reviewer: There probably should be a reference for the sentence ending on line 55.
Authors: We thank the referee for his/her suggestion. An appropriate reference has been added.
6) Reviewer: Line 57 “Dynein has been reported….” Is outside the subject of the paragraph.
Authors: Thank you for your remark. In the revised version the sentence has been removed.
7) Reviewer: Line 57 process to processes.
Authors: In the revised version the sentence has been modified.
8) Reviewer: Reference for sentence ending on line 115.
Authors: We thank the referee for his/her suggestion. In the revised version of the paper a proper reference has been added.
9) Reviewer: The sentence starting on line 119 is confusing due to the parathesis (missing one); the next sentence is outside the subject of the paragraph, and it is unclear why the authors focused on MFN2 (why not OPA1?).
Authors: We apologize for this typing error, which, in the revised version of the paper, has been corrected. The next sentence, as suggested, has been removed.
10) Reviewer: Line 134, “MFF appears to be the mitochondrial receptor for DRP1.” It is one of the receptors, along with Fis1 (should probable cite Loson et al. Mol Biol Cell. 2013).
Authors: We thank the referee for his/her suggestion. In the revised version of the paper the sentence has been corrected and the suggested reference has been added.
11) Reviewer: Line 194, clearance damaged to clear damaged.
Authors: In the revised version the error has been corrected.
12) Reviewer: Line 201, from cytosol to from the cytosol.
Authors: In the revised version the sentence has been corrected.
13) Reviewer: Line 212, OMM introduced for outer mitochondrial membrane was also introduced on line 95 but not used consistently throughout the manuscript.
Authors: We thank the referee for his/her observation. In the revised version this inaccuracy has been eliminated.
14) Reviewer: Line 337, several experimental evidence to experimental evidence
Authors: In the revised version of the paper the sentence has been corrected.
15) Reviewer: Lines 448-449, “or to be” to or be
Authors: In the revised version of the paper the error has been removed.
16) Reviewer: There are examples when mechanistic information is stated without describing its overall significance/relevance. For example, what is the significance of perinuclear mitophagy in lymphomas? Which process at the leading edge of migrating cells require energy? Current knowledge of mechanisms controlling anterograde versus retrograde mitochondrial transport?
Authors: We thank the referee for his/her suggestion. In the revised version the sentences highlighted have been discussed in more detail in Sections 4 and 5. As regards the current knowledge of the mechanisms that control antegrade versus retrograde mitochondrial transport, we mentioned it in Session 5, in which the motor and adaptive proteins involved are cited. The specific distinction between the two mechanisms is beyond the scope of this paper.
Reviewer 2 Report
I have read with interest the review paper entitled “Microtubules-based mitochondria dynamics as a valuable therapeutic target in cancer“ and I believe that it summarizes well the current state of the art and it includes nice illustrative Figures and is worth publishing.
- Figure 1- the symbol of the scale should symbolize the equity between the two sides, but in the current state it implies that cancer cells are more proliferative and invade and metastasize while apoptosis resistance is less important. That is not the real description, I would show the symbol at balance e.g. both phenotype can contribute equally and i would also include metabolic rewiring/adaptation
- Figure 3 panel C „SNPH ancor“ should be „SNHP anchor“
- Some important published papers are not cited, although the topic is extensively discussed, for example the Cell metab paper by Tan et al. and recently Bajzikova et al. are not listed although they are directly showing the horizontal transfer of mitochondria and the crucial role of DHODH enzyme in pyrimidine synthesis that depends on functional OXPHOS while mitochondrial ATP generation is dispensable similarly, current Nature paper by Heindrichs et al., describing the nature of the mitochondrial movement using TRAK1, is not listed. Please consider adding these sources (doi: 10.1016/j.cmet.2014.12.003.; doi: 10.1038/s41467-020-16972-5)
- On the same note, when the authors discuss the role of mitochondrial fission and the movement of mitochondria towards the cellular periphery as a phenotype connected with migratory and metastatic phenotype, they should include citation of paper by Tomkova et al., where authors describe fragmentation of mitochondria, phosphorylation of DRP1 and movement of fragmented mitochondria towards the periphery in the model of tamoxifen-resistant cells (doi: 1016/j.freeradbiomed.2019.09.004). In general, the connection and role of ROS and its connection with mitochondrial dynamics could be reviewed a bit more as it could be very important modulator of mitochondrial dynamics.
- The connection between mitophagy and and anti-tumour immunity should also discussed as it could be very relevant also in the future use in clinics given the growing body of evidence suipporting the active role of immune system in anti-tumor response and the success of immunomodulation in certain types of cancer(doi: 1016/j.cell.2018.05.028)
Author Response
Replies to the comments
Reviewer #2
We thank this reviewer for his/her comments and suggestions that have helped us improve our work.
1) Reviewer: I recommend doing a revision of the English language to avoid the repetition of terms and use more appropriate scientific terms / words (for example generators. Important concepts, pathways and therapeutic ideas are underdeveloped in most of the paragraphs. The authors analyze in detail the main concepts related to the mitophagy pathway and its regulation; It also refers to a large number of works. However, the concept of mitophagy or its inclusion / function within the mitochondrial dynamics is not reflected in the simple and extended abstracts.
Authors: We thank this reviewer for reading our work carefully and for helpful advice on implementing the quality of this paper. As recommended we did a revision of the English and removed the repetitions of the terms present through the paper. In addiotn, we modified either the simple or extended abstract as suggested.
2) Reviewer: I recommend that the authors 1) really justify that the fourth section entitled "mitophagy" provides essential content to the review 2) if they do, they should reference it correctly in the abstract so that the reader knows the concepts that the review is going to address. If the editor considers it, section 4 could be removed from the review by extending and appearing as a section in no man's land.
Authors: We thank the referee for his/her observations. In the revised version of the paper, the section entitled "Mitophagy" has been deeply modified and better contextualized to the content to the review.
3) Reviewer: The paragraph between lines 166 and 184 of the third section and entitled Mitochondrial fission and fusion, must be rewritten and explained in a more coherent way.
Authors: We particularly appreciated this suggestion. We have completely rewritten the paragraph making it more flowing and clear.
4) Reviewer: Figure 1, lines 190 and 191, must be expanded so that the reader can correctly follow the order of the figures that appear. the term "cartoon" is not appropriate for a scientific figure. Furthermore, I consider that figure 1 is not reflected in the main text of this review, referring to section 3 "mitochondrial fission and fusion". Authors should correct these details.
Authors: We completely agree with the criticisms raised by the reviewer. The legend of Figure 1 (now become Figure 2) has been enlarged to allow the reader to easily follow the content of the figure itself. In the revised version of the paper the term "cartoon" has been modified, and a new Figure describing the general contents of this review has been inserted as a new Figure 1.
5) Reviewer: The manuscript should emphasize much more in the therapeutic message that reflects the title of the review. Section 8 should be expanded and duly documented, even to the detriment of section 4 (mitophagy). Authors should consider this contribution, due to the attractive message they send in title and abstracts.
Authors: We thank the referee for his/her suggestions. In the new version of the paper the Section 8 (Mitochondrial Dynamics and Cancer Therapy) has been implemented on the basis of literature data.
Reviewer 3 Report
General Comments:
- In general, the manuscript is not well developed. First I recommend doing a revision of the English language to avoid the repetition of terms
and use more appropriate scientific terms / words (for example generators, . Important concepts, pathways and therapeutic ideas are underdeveloped in most of the paragraphs. The authors analyze in detail the main concepts related to the mitophagy pathway and its regulation; It also refers to a large number of works. However, the concept of mitophagy or its inclusion / function within the mitochondrial dynamics is not reflected in the simple and extended abstracts. - I recommend that the authors 1) really justify that the fourth section entitled "mitophagy" provides essential content to the review 2) if they do, they should reference it correctly in the abstract so that the reader knows the concepts that the review is going to address. If the editor considers it, section 4 could be removed from the review by extending and appearing as a section in no man's land
- The paragraph between lines 166 and 184 of the third section and entitled Mitochondrial fission and fusion, must be rewritten and explained in a more coherent way. However, it is well referenced.
Minor considerations :
- Figure 1, lines 190 and 191, must be expanded so that the reader can correctly follow the order of the figures that appear. the term "cartoon" is not appropriate for a scientific figure. Furthermore, I consider that figure 1 is not reflected in the main text of this review, referring to section 3 "mitochondrial fission and fusion". Authors should correct these details.
The manuscript should emphasize much more in the therapeutic message that reflects the title of the review. Section 8 should be expanded and duly documented, even to the detriment of section 4 (mitophagy). Authors should consider this contribution, due to the attractive message they send in title and abstracts.
Author Response
Replies to the comments
Reviewer #3
We thank this reviewer for his/her comments and suggestions that have helped us improve our work.
1) Reviewer: Figure 1- the symbol of the scale should symbolize the equity between the two sides, but in the current state it implies that cancer cells are more proliferative and invade and metastasize while apoptosis resistance is less important. That is not the real description, I would show the symbol at balance e.g. both phenotype can contribute equally and i would also include metabolic rewiring/adaptation.
Authors: We thank the referee for his/her useful suggestions. In the revised version of the paper we have modified Figure 1 (now become Figure 2) following the referee's instructions.
2) Reviewer: Figure 3 panel C, SNPH ancor“ should be „SNHP anchor“.
Authors: We apologize for this typing error, which has been corrected in the revised version of the Figure 3 (now Figure 4).
3) Reviewer: Some important published papers are not cited, although the topic is extensively discussed, for example the Cell metab paper by Tan et al. and recently Bajzikova et al. are not listed although they are directly showing the horizontal transfer of mitochondria and the crucial role of DHODH enzyme in pyrimidine synthesis that depends on functional OXPHOS while mitochondrial ATP generation is dispensable similarly, current Nature paper by Heindrichs et al., describing the nature of the mitochondrial movement using TRAK1, is not listed. Please consider adding these sources.
Authors: We thank the referee for his/her suggestions. In the revised version of the paper all the references indicated have been added.
4) Reviewer: On the same note, when the authors discuss the role of mitochondrial fission and the movement of mitochondria towards the cellular periphery as a phenotype connected with migratory and metastatic phenotype, they should include citation of paper by Tomkova et al., where authors describe fragmentation of mitochondria, phosphorylation of DRP1 and movement of fragmented mitochondria towards the periphery in the model of tamoxifen-resistant cells. In general, the connection and role of ROS and its connection with mitochondrial dynamics could be reviewed a bit more as it could be very important modulator of mitochondrial dynamics.
Authors: We thank the referee for his/her suggestions. In the revised version of the paper the reference indicated has been added and the role of ROS and its connection with mitochondrial dynamics has been amplified.
5) Reviewer: The connection between mitophagy and anti-tumor immunity should also discussed as it could be very relevant also in the future use in clinics given the growing body of evidence supporting the active role of immune system in anti-tumor response and the success of immunomodulation in certain types of cancer.
Authors: We thank the referee for his/her suggestion. In the revised version the connection between mitophagy and and anti-tumour immunity has been discussed in the section 4.
Round 2
Reviewer 1 Report
All concerns were addressing, and the manuscript has all aspects of a useful review. There are a few minor typos in the modified figure legends that will probably be identified during proofing of the manuscript.